



# ChemicalDrift 1.0: an open-source Lagrangian chemical fate and transport model for organic aquatic pollutants

Manuel Aghito[1,4], Loris Calgaro[2], Knut-Frode Dagestad[1], Christian Ferrarin[3], Antonio Marcomini[2], Øyvind Breivik[1,4], and Lars Robert Hole[1]

[1]The Norwegian Meteorological Institute, Bergen, Norway
[2]Department of Environmental Sciences, Informatics and Statistics, University Ca' Foscari of Venice, Venice, Italy
[3]CNR - National Research Council of Italy, ISMAR - Institute of Marine Sciences, Venice, Italy
[4]University of Bergen, Bergen, Norway

**Correspondence:** Manuel Aghito (manuel@met.no)

**Abstract.** A new model for transport and fate of chemicals in the aquatic environment is presented. The tool, named ChemicalDrift, is integrated in the open-source Lagrangian framework OpenDrift, and is hereby presented for organic compounds. The supported chemical processes include the degradation, the volatilization, and the partitioning between the different phases that a target chemical can be associated to in the aquatic environment, e.g. dissolved, bound to suspended particles, or deposited

to the seabed sediments. The dependencies of the chemical processes on changes of temperature, salinity, and particle concentration, are formulated and implemented. The chemical fate modelling is combined with wide support for hydrodynamics by the integration within the Lagrangian framework which provides, e.g., advection by ocean currents, diffusion, wind induced turbulent mixing, and Stokes drift generated by waves. A powerful interface to a wide range of available metocean data is made available by the integration, making the tool flexible and adaptable to different spatio-temporal scales and fit for modelling of

complex coastal regions. Further inherent capabilities of the Lagrangian approach include the seamless tracking and separation of multiple sources as, e.g., pollutants emitted from ships or from rivers or water treatment plants. Specific interfaces to a dataset produced by a model of emissions from shipping, and to an unstructured-grid oceanographic model of the Adriatic Sea, are provided. The model includes a database of chemical parameters for a set of poly-aromatic hydrocarbons, and a database of emission factors for different chemicals found in discharged waters from sulphur emission abatements systems in marine

vessels. A post-processing tool for generating mean concentrations of a target chemical, over customizable spatio-temporal grids, is provided. Model development and functional testing are presented, while tuning of parameters, validation, and reporting of numerical results, are planned as future activities. The ChemicalDrift model flexibility, functionalities, and potential, are demonstrated through a selection of examples, introducing the model as a valuable tool for chemical fate and transport, that can be applied to assessment of the risks of contamination by organic pollutants in the aquatic environment.

# 1   Introduction

The negative effects of chemical pollution on the environment and the human health have been long established (Naidu et al., 2021), but only more recently has the need for integrated strategies and solutions for the simultaneous management of multiple



stressors been widely recognized (Pirotta et al., 2022). In fact, one of the main challenges in assessing the impact of anthropogenic stressors on the environment, and especially on aquatic ecosystems, is to estimate the exposure to these contaminants
while also keeping track of their origin, in particular when multiple sources of chemical pollution are present.

Lagrangian models offer the possibility to investigate the transport of chemicals without losing information on their origin, thus facilitating the assessment of the contribution each source has on the overall exposure of the target ecosystem. Furthermore, since their spatial resolution is not inherently limited to the resolution of Eulerian grid boxes and no grid generation is required (Azarpira et al., 2021), Lagrangian models are suitable to study areas with complex geometries such as coasts, lagoons and
archipelagos. The capabilities of these models to predict the transport of chemicals and particles emitted into the atmosphere by a variety of sources (e.g., industrial installations (Hirdman et al., 2010; Lee et al., 2014), nuclear accidents (Becker et al., 2007; Draxler et al., 2015), and volcanic eruptions (Prata et al., 2007; D'Amours et al., 2010)) have been widely studied (Onink et al., 2021). On the other hand, the potential of Lagrangian models applied to the fate and transport of organic chemicals in the water compartment is not fully investigated.

It is recognized that coastal environments are of particular concern due the presence of multiple stressors (e.g., wastewater discharge, soil and sediment contamination, agricultural and municipal run-off, and combustion of fossil fuels by civil and industrial activities) (Danovaro and Boero, 2019; Szymczycha et al., 2019; Calgaro et al., 2019, 2021) posing risk to the ecosystem and human health due to water and air contamination. An additional specific stressor is represented by emissions of $SO_x$, $NO_x$ and other air-borne contaminants (e.g., particulate matter and polyaromatic hydrocarbons) from shipping activity,
which has been shown to significantly degrade air quality (Zhang et al., 2021), especially in areas characterized by high traffic such as ports and shipping lanes. For these reasons, the International Maritime Organization (IMO), starting from 1 January 2020, has introduced new global regulations limiting the sulphur content of fossil fuels to a maximum of 0.50 % weight/weight (w/w), unless they are fitted with a suitable exhaust gas cleaning system ("scrubbers") (Vedachalam et al., 2022). Furthermore, the more stringent limit of 0.10 % w/w sulphur has been applied to especially sensitive areas (i.e., Emission Control Areas,
ECA) like the Baltic Sea, the North Sea, and the USA coasts (Chang et al., 2018). Plans to include other areas into ECA's list, like the Mediterranean Sea, are under discussion (Testa, 2020).

The scrubbing process is implemented by leading the exhaust gas through a fine water mist where $SO_x$ and $NO_x$ are dissolved together with particulate matter, metals, and organic compounds (Lunde Hermansson et al., 2021), thus large volumes of toxic (Turner et al., 2017) and highly acidified effluents (Turner et al., 2018) (where several pollutants as, e.g., Cu, Pb, Hg, V,
Zn, As, Ni, and polyaromatic hydrocarbons (PAHs) have been detected) are produced and discharged into the sea. Moreover, due to the increase of shipping traffic predicted for the next decades (Sardain et al., 2019), it is crucial to develop methods and tools to investigate the impact of scrubber use on the marine ecosystem in relation to other sources of chemical pollution.

The present work is carried out under the scope of the Horizon 2020 EMERGE project, which aims to develop an integrated modelling framework to assess the combined impact of shipping emission control options on the aquatic and atmospheric
environment. Utilizing the open-source Lagrangian framework OpenDrift (Dagestad et al., 2018), a new model for chemical transport and fate of aquatic pollutants, named ChemicalDrift, has been developed and is hereby presented and demonstrated.





This article is organized as follows: the relevant chemical processes implemented in ChemicalDrift are presented and formulated in Sect. 2; the set of inputs utilized in this work, including metocean forcing data, chemical parameters for the studied pollutants, and emission source data, are presented in Sect. 3, and a selection of demonstrative examples are presented in Sect. 4. Planned future developments and conclusions are summarized in Sect. 5.

## 2 Chemical fate modelling

The new module ChemicalDrift is presented in this section supported by a detailed description of the modelled chemical processes. The module is coded in Python, and integrated in the open-source Lagrangian framework OpenDrift (Dagestad et al., 2018) where the fundamental physical processes of advection and diffusion as well as atmospheric and oceanographic forcing were previously implemented. While OpenDrift already contains a module for oil pollution, OpenOil (Dagestad et al., 2018), which has validated and used for simulations of severe oil spills (Röhrs et al., 2018; Brekke et al., 2021; Hole et al., 2021)), the new module presented here is intended for simulations of dispersion of contaminants in much lower concentrations in the sea water. Only the chemistry of non-ionizable organic compounds is described in this work, since the chemistry of metals have been already implemented in the Radionuclides OpenDrift module (Simonsen et al., 2019a) which is integrated in ChemicalDrift.

The reactions implemented in ChemicalDrift include a partitioning scheme between the different phases that a target chemical can be associated to within the aquatic environment, the degradation of organic chemicals in the water column and in the sediments, and the volatilization of dissolved chemicals from the water to the atmosphere. The dependencies of the each reaction on temperature and salinity changes are described and implemented, based on open scientific literature. Sedimentation and resuspension are also presented in this section since these physical processes are strongly related to the chemical phase partitioning.

### 2.1 Dynamic Partitioning between compartments

Chemicals in the aquatic environment can be associated to different components of this medium (i.e., dissolved in the water or bound to dissolved organic carbon (DOC), suspended particle matter (SPM) or sediments) and thereby exposed to different physical and chemical processes. For example, dissolved chemicals will be transported by advection due to water currents, and by turbulent diffusion processes, such as wind-induced vertical mixing, while chemicals adsorbed to solid particles will also be affected by gravity and might sink towards the seafloor. Furthermore, the different distribution between dissolved and bound chemicals will also influence its availability to degradation processes like hydrolysis, biodegradation and photolysis. In computational models it is therefore crucial to process each of these components in distinct compartments.

In ChemicalDrift, each Lagrangian element represents a given mass of a target chemical, and the partitioning between the media components is implemented by assigning each element to a single corresponding model compartment. The partitioning is carried out dynamically, as opposed to a simpler steady-state approach, and transfer rates between compartments are calculated





and applied at each time step. The dynamic approach is fit for the changing environmental conditions each Lagrangian element is exposed to (i.e. water temperature, salinity, SPM and DOC concentration) as it is transported in the media.

The dynamic partitioning scheme used in ChemicalDrift was first introduced by Periáñez and Elliott (2002) and implemented in the OpenDrift Radionuclides module (Simonsen et al., 2019a, b), where it was applied to metals. In this work, several adaptations were made to expand its applicability to organic chemicals and consider the influence of temperature and salinity changes on the transfer rates.

The main phases considered in the algorithm (Fig. 1) include: (i) dissolved in the water fraction, (ii) adsorbed to SPM
fraction, and (iii) sediment fraction. The sediment fraction includes both the chemical adsorbed to solid particles that have settled to the seafloor, and the chemical dissolved in the interstitial water present in the space between the settled particles. Optional model compartments can also be activated. The chemical can also be modelled as adsorbed to DOC and form other aggregates that will be subjected to sinking (i.e., the DOC phase). Adsorption to solid particles can be treated as a two-step process where the diffusion of the chemical within the particle is also considered (i.e., a slowly reversible fraction, SR) for both
SPM and sediments. Finally, an optional compartment for buried sediments is implemented, representing a sink for the target chemical that is progressively buried below layers of overlying sediments and is therefore not available for exchanges with the water column.

### 2.1.1   Adsorption and desorption rates

Transfer rates are calculated adapting the equations described by Simonsen et al. (2019a) and applied using the approach
explained in Sect. 2.1.2 for estimating the adsorption and desorption kinetic constants, $k_{ads}$ [L kg$^{-1}$ h$^{-1}$] and $k_{des}$ [h$^{-1}$].

Transfer rates from the dissolved phase to the DOC, SPM and sediments compartments (i.e., $k_{12}$, $k_{13}$, and $k_{14}$) and the corresponding desorption rates (i.e. $k_{21}$, $k_{31}$, and $k_{41}$) are given by

$$k_{12} = k_{ads} C_{DOC},$$
$$k_{13} = k_{ads} C_{SPM},$$
$$k_{14} = k_{ads} \rho_{sed} (1 - p_{sed}) L \phi \delta / H, \tag{1}$$

$$k_{21} = k_{des} = k_{ads} / K_{d,DOC},$$
$$k_{31} = k_{des} = k_{ads} / K_{d,SPM},$$
$$k_{41} = \phi k_{des} = \phi k_{ads} / K_{d,SPM}. \tag{2}$$

Here the indices refer to the numbering illustrated in Fig. 1; $C_{DOC}$ is the concentration of DOC in the water column [kg L$^{-1}$], $C_{SPM}$ is the concentration of SPM in the water column [kg L$^{-1}$]; $\rho_{sed}$ is the density of sediment solids [kg m$^{-3}$]; $p_{sed}$ is the porosity of the sediments; $L$ [m] is thickness of the sediment layer which interact with the water column; $H$ [m] gives the thickness of the water layer above the sea bottom that interacts with the sediment; $\phi$ is the fraction of effective
sorbents considering that a fraction of sediment particle surface may be unavailable for sorption/desorption interactions due to





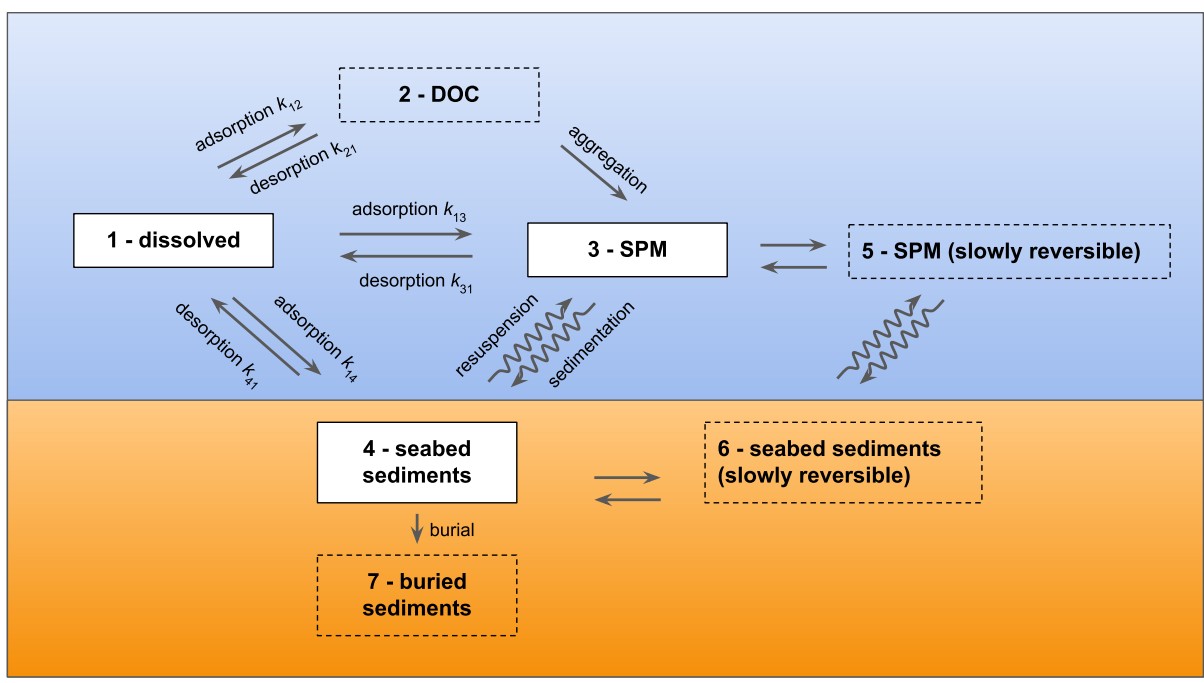

**Figure 1.** The model compartments implemented in ChemicalDrift and the corresponding exchange processes and transfer rates, adapted from Simonsen et al. (2019a). The default (dissolved, adsorbed to SPM and sediments) compartments are indicated with solid lines. Optional phases are depicted with dashed lines.

neighbouring sediment particles; $\delta$ is logical variable which ensures that only the chemicals in the seabed interaction layer are able to adsorb to sediments and is set to 1 when the distance to seabed is less than $H$, and 0 elsewhere. Default values or typical ranges are summarized in Table 1. The concentrations $C_{\mathrm{DOC}}$ and $C_{\mathrm{SPM}}$ are provided as modelled or experimentally derived data fields. If only 2-D surface data is available, the vertical profiles are estimated assuming a constant concentration in the
mixed layer (equal to the surface values) and exponentially decreasing with depth below the mixed layer, with user definable constants, in accordance with the observations of biogeochemical argo floats measurements reported by Galí et al. (2022).

    The stochastic method for calculating the probability of phase change between model compartments given the time step and the transfer rates is described by Periáñez and Elliott (2002).

### 2.1.2   General estimation of $k_{\mathrm{ads}}$ and $k_{\mathrm{des}}$ for organic chemicals

The partitioning of organic chemicals to organic carbon has been reported as involving by rapid adsorption to particle surfaces, followed by slow movement into, and out of, organic matter and porous aggregates (Karickhoff and Morris, 1985), thus determining the time needed for the attainment of equilibrium (Park and Clough, 2014).





**Table 1.** Parameters for the calculation of sediment layer adsorption and desorption rates.

| Symbol | Parameter | Value / Range | Unit | Reference |
|---|---|---|---|---|
| $\rho_{\mathrm{sed}}$ | density of sediment solids | 2.33 | kg L$^{-1}$ | Fantke (2018); Honti et al. (2016); Kettelarij et al. (2019) |
| $p_{\mathrm{sed}}$ | porosity of sediment layer | 0.8 | - | Paterson and Mackay (1995); Kettelarij et al. (2019) |
| $L$ | thickness of sediment active layer | $0.03 - 0.10$ | m | Periáñez (2012) |
| $H$ | thickness of water column interaction layer | $0.2 - 3$ | m | Kobayashi et al. (2007); Periáñez and Elliott (2002) |
| $\phi$ | fraction of effective sorbents | $0.01 - 0.1$ | - | Periáñez (2008); Periáñez et al. (2013) |

The solid-water partition coefficient $K_{\mathrm{d}}$ [L kg$^{-1}$] is the ratio between the adsorption and desorption kinetic constants

$$K_{\mathrm{d}} = k_{\mathrm{ads}}/k_{\mathrm{des}}. \tag{3}$$

130 Given this context, Karickhoff and Morris (Karickhoff and Morris, 1985; Site, 2001) found the reciprocal of $k_{\mathrm{des}}$ to be a linear function of $K_{\mathrm{d}}$ over three orders of magnitude ($r^2 = 0.87$), following $1/k_{\mathrm{des}} \approx 0.03 K_{\mathrm{d}}$. Using this approximation and the definition of $K_{\mathrm{d}}$, the adsorption and desorption kinetic constants can be estimated as

$$k_{\mathrm{ads}} \approx 33.3 \text{ L kg}^{-1}\text{h}^{-1},$$
$$k_{\mathrm{des}} = k_{\mathrm{ads}}/K_{\mathrm{d}} \approx 33.3/K_{\mathrm{d}} \text{ h}^{-1}. \tag{4}$$

The value of $K_{\mathrm{d}}$ for a given chemical in the aquatic environment can be subjected to significant changes due to, e.g., tempera-

135 ture, salinity, SPM and DOC concentration variations, and pH, therefore the use of experimental local measurements of $K_{\mathrm{d}}$ is recommended. However, if such data is not available the model will estimate the $K_{\mathrm{d}}$ as described below.

For organic chemicals it is assumed that the organic carbon fraction of solid particles (i.e., suspended particle matter and sediments) is almost entirely responsible for their sorbing capacity, and therefore $K_{\mathrm{d}}$ can be expressed as the product of a water-organic carbon partition coefficient (i.e. $K_{\mathrm{OC}}$, [L kg$_{\mathrm{OC}}$$^{-1}$]) and the organic carbon content of the dry matter ($f_{\mathrm{OC}}$, [g$_{\mathrm{OC}}$

140 g$^{-1}$]).

$$K_{\mathrm{d}} = K_{\mathrm{OC}} f_{\mathrm{OC}} \tag{5}$$

Typical values for $f_{\mathrm{OC}}$ are in the range $0.01 - 0.10$ g$_{\mathrm{OC}}$ g$^{-1}$ (Seiter et al., 2004). Relationships between $K_{\mathrm{OC}}$ and the octanol-water partition coefficient $K_{\mathrm{OW}}$ [L kg$^{-1}$] are available from previous studies. In the case of non-ionizing organic chemicals, the following formulations reported by Park and Clough (2014) are used to estimate $K_{\mathrm{OC}}$ for SPM and DOC,

145 respectively

$$K_{\mathrm{OC,DOC}} \approx 2.88(K_{\mathrm{OW}})^{0.67} \tag{6}$$

$$K_{\mathrm{OC,SPM}} \approx 2.62(K_{\mathrm{OW}})^{0.82} \tag{7}$$





### 2.1.3 Effect of temperature and salinity on the solid-water partition coefficient

Experimental values of $K_{\mathrm{OC}}$ and $K_{\mathrm{d}}$ are usually obtained at standard or ambient temperature and for fresh water, but temperature and salinity variations have been shown to have a strong impact on the adsorption of chemicals to SPM, DOC, and sediments. The following equation and correction factors are applied to obtain to obtain the solid-water partition coefficient $K_{\mathrm{d}}(T, S)$ for a desired value of temperature $T$ [°K] and salinity $S$ [PSU] based on a known reference value $K_{\mathrm{d}}(T_{\mathrm{ref}}, 0)$:

$$K_{\mathrm{d}}(T, S) = K_{\mathrm{d}}(T_{\mathrm{ref}}, 0)\, \mathrm{TempCorr}(T, T_{\mathrm{ref}}, \Delta H)\, \mathrm{SalinityCorr}_{K_d}(S) \tag{8}$$

$$\mathrm{SalinityCorr}_{K_d}(S) = 10^{k_{\mathrm{S}}\, C_{\mathrm{salt}}(S)} \tag{9}$$

$$\mathrm{TempCorr}(T, T_{\mathrm{ref}}, \Delta H) = e^{-\frac{\Delta H}{R}\left(\frac{1}{T} - \frac{1}{T_{\mathrm{ref}}}\right)} \tag{10}$$

where $k_{\mathrm{S}}$ is the Setschenow (salting out) constant in seawater of the non-ionizable organic chemical with respect to the specific salt considered [L mol$^{-1}$], and $C_{\mathrm{salt}}$ is the molar concentration of the salts in solution [mol L$^{-1}$]. Salt concentration is calculated starting from salinity considering the molecular weight of sea water salt, $\mathrm{MolWt}_{\mathrm{salt}} = 68.35$ g mol$^{-1}$ (Schwarzen-
bach et al., 2016), and the water density, $\rho$.

$$C_{\mathrm{salt}} = \left(\frac{S}{\mathrm{MolWt}_{\mathrm{salt}}}\right) \rho \tag{11}$$

The correction factor for temperature is based on the Arrhenius equation where $\Delta H$ [J mol$^{-1}$] is the enthalpy of adsorption for the given chemical and R is the universal gas constant ($R = 8.3145$ J mol$^{-1}$ K$^{-1}$). The model utilize distinct enthalpies for DOC and SPM, indicated as $\Delta H_{\mathrm{KOC,DOC}}$ and $\Delta H_{\mathrm{KOC,SPM}}$, respectively.

### 2.1.4 Sedimentation and resuspension

Particles sedimentation and resuspension dynamics in ChemicalDrift is partly based on methods previously implemented in other OpenDrift modules. The vertical motion of Lagrangian elements representing dissolved chemicals, or chemicals adsorbed to DOC, is calculated as for a passive tracer, by adding vertical currents (typically neglected in operational ocean models) to the effect of wind induced turbulent diffusion.

Lagrangian elements associated to SPM are also affected by gravity and will sink towards the seafloor, hence a third component has to be added to calculate the vertical motion. This term is referred to as the terminal velocity, and is calculated from Stokes' Law (Stokes, 1851), assuming the particles to be spherical with known density and diameter, and low Reynolds numbers. Empirical relationships are utilized to update the water density (Fofonoff and Millard Jr, 1983) and viscosity (Ådlandsvik, 2000; Myksvoll et al., 2013) based on $T$ and $S$ as the elements sink.





When sinking particles reach the sea floor the chemical elements are transferred to the sediment layer compartments, and thus can be subjected to either sediment burial or resuspension. Resuspension occurs when the horizontal current is greater than a given threshold $v_{\mathrm{crit}}$. Resuspended elements are lifted from the previous seafloor depth $z_{\min}$(depth below sea level, negative) to a depth $z_{\min} + z_{\mathrm{res}} + \mathcal{N}(z_{0,\mathrm{res,unc}})$, where $z_{\mathrm{res}}$ is the resuspension depth, $z_{\mathrm{res,unc}}$ is the resuspension depth uncertainty, and $\mathcal{N}(0, z_{\mathrm{res,unc}})$ is a randomly generated number from a normal distribution with zero mean and standard deviation $z_{\mathrm{res,unc}}$.

## 2.2   Degradation

Organic chemicals in the aquatic environment can be degraded due to various processes, such as biodegradation, photodegradation, and hydrolysis. Modelling each of these reactions separately requires a large amount of information on the environmental behavior of the target chemical and on the characteristics of the selected study area. Since these data are typically difficult to obtain with sufficient accuracy for a wide range of chemicals and case study regions, a simpler approach is implemented in the current version of ChemicalDrift: degradation is implemented considering distinct overall reaction rates for the water column (subscript Wtot) and the sediments (subscript Stot), and modelled as a first order kinetic decay of the mass associated with each Lagrangian element; in detail, overall degradation rate constants $k$, or the corresponding half-lives $t_{1/2}$, can be obtained experimentally without separating the effects of the three sub-processes. The mass $m_i(t)$ of each Lagrangian chemical element is calculated at each time step applying

$$
m_i(t + \Delta t) = \begin{cases} m_i(t)e^{-k_{\mathrm{Wtot}}\Delta t}, & \text{if } i \in (\mathrm{Dissolved}, \mathrm{DOC}) \\ m_i(t)e^{-k_{\mathrm{Stot}}\Delta t}, & \text{if } i \in (\mathrm{Sediments}) \\ m_i(t), & \text{otherwise} \end{cases} \tag{12}
$$

where $k_{\mathrm{Wtot}}$ is the overall (total) degradation constant for the chemical either dissolved in the water column or adsorbed to dissolved organic carbon/matter, $k_{\mathrm{Stot}}$ is the overall degradation constant for chemicals in the sediment layer, and $\Delta t$ is the time step interval. As reported by among others Kettelarij et al. (2019), it is assumed that degradation of chemicals adsorbed to suspended particles can be neglected. Degradation rate constants can be specified directly or calculated from the corresponding half-life constants $t_{1/2\mathrm{Wtot}}, t_{1/2\mathrm{Stot}}$ applying $k = -\log(0.5)/t_{1/2}$.

Furthermore, the effect of temperature is considered by applying to each degradation rate constant a correction factor calculated by applying the Arrhenius equation as in Eq. (10) using the reference temperature (i.e., $T_{\mathrm{refWtot}}$, and $T_{\mathrm{refStot}}$) and enthalpy (i.e., $\Delta H_{\mathrm{Wtot}}$ and $\Delta H_{\mathrm{Stot}}$) associated with each process. Salinity has also been demonstrated to have an impact on the microbial populations responsible for the biodegradation of organic chemicals, but since the modelling of these effects would require the integration of ChemicalDrift with a biogechemical model, is has been considered outside the scope of this work and neglected (Park and Clough, 2014).





### 2.3 Volatilization

Volatilization is modelled using the "stagnant boundary theory", or two-film model, in which the target chemical must diffuses across both a stagnant water layer and a stagnant air layer to volatilize out of the water column (Schwarzenbach et al., 2016;

Parnis and Mackay, 2020; Park and Clough, 2014; Kettelarij et al., 2019).

According to the literature (Schwarzenbach et al., 2016), the volatilization rate [kg m$^{-3}$ s$^{-1}$] can be calculated based on the properties of the two-film interface and the concentrations of the chemical,

$$\text{volatilization} = \frac{\partial}{\partial t} C_{\text{water}} = -(\text{MTC}_{\text{vol}}/H_{\text{MLD}})(C_{\text{sat}} - C_{\text{water}}). \tag{13}$$

Here, $C_{\text{water}}$ [kg m$^{-3}$] is the concentration of the chemical in the water phase, $C_{\text{sat}}$ [kg m$^{-3}$] is the saturation concentration of

the pollutant in equilibrium with the gas phase, $H_{\text{MLD}}$ [m] is the thickness of the mixed layer if the water column is stratified or the maximum depth otherwise, and $\text{MTC}_{\text{vol}}$ [m s$^{-1}$] is the overall volatilization mass transfer coefficient from water to air.

If pollutants can be assumed to have negligible concentration in air (i.e, $C_{\text{sat}} \approx 0$), then the equation simplifies to a first order problem (i.e., $\frac{\partial}{\partial t} C_{\text{water}} = k_{\text{vol}} C_{\text{water}}$) that is solved by $C_{\text{water}}(t_2) = C_{\text{water}}(t_1) e^{-k_{\text{vol}}(t_2 - t_1)}$ where the volatilization rate constant (s$^{-1}$) can be calculated from Eq. (14).


$$k_{\text{vol}} = \text{MTC}_{\text{vol}}/H_{\text{MLD}} \tag{14}$$

This assumption is often considered acceptable and is widely applied for chemical fate modelling (Park and Clough, 2014; Deltares, 2014), especially in the case of chemicals with very low volatility or that can be found mostly bound to suspended particles in the air, such as the 5- and 6-rings PAHs, but can be problematic for more volatile chemicals (e.g., Naphthalene)

that are found in significant concentrations in the gas phase. In those cases, a coupled atmospheric-ocean model is required for accurate calculation of exchanges across the interface, but ChemicalDrift can offer a reasonable compromise between accuracy of the results and resources needed.

In the Lagrangian framework utilized by the proposed model concentrations are not calculated at each time step, hence the equivalent effect is obtained applying an exponential decay to the mass $m_i(t)$ of all dissolved elements $i$ within the mixed

layer, as expressed by $m_i(t + \Delta t) = m_i(t) e^{-k_{\text{vol}} \Delta t}$.

As reported above, $k_{\text{vol}}$ can be estimated starting from $\text{MTC}_{\text{vol}}$, whose reciprocal can be interpreted as the total resistance to the mass transfer through the two-film interface and therefore can be expressed as the sum of the resistances of each layer (Eq. 15),

$$\frac{1}{\text{MTC}_{\text{vol}}} = \frac{1}{\text{MTC}_{\text{w}}} + \frac{1}{\text{MTC}_{\text{a}} H^{\text{cc}}}. \tag{15}$$

Here, $\text{MTC}_{\text{a}}$ and $\text{MTC}_{\text{w}}$ [m s$^{-1}$] are the air-side and water-side mass transfer coefficients, respectively, $H^{\text{cc}}$ is the dimensionless Henry's law constant.

The diffusion rates of a chemical in these stagnant boundary layers can be related to the known diffusion rates of reference substances such as oxygen and water vapor (Schwarzenbach et al., 2016). Based on the *Boundary Layer Theory* (Deacon, 1977)





the mass transfer coefficient for a target chemical can be obtained from these by applying a correction factor that depends on

the ratios of the Schmidt numbers of the target and reference chemicals. The following relations proposed by (McGillis et al., 2001) and (Johnson, 2010) and summarized by (Schwarzenbach et al., 2016) are used in this work:

$$
\mathrm{MTC_a} = 0.01 \left( \frac{\mathrm{MolWt_{H_2O}}}{\mathrm{MolWt}} \right)^{1/3} \mathrm{MTC_{H_2Oa}}
$$

$$
\mathrm{MTC_{H_2Oa}} = 0.1 + \frac{U_{10}(6.1 + 0.63 U_{10})^{1/2}}{13.3(\mathrm{Sc_{H2Oa}})^{1/2} + (6.1 \times 10^{-4} + 6.3 \times 10^{-5} U_{10})^{-1/2} - 5 + 1.25 \ln(\mathrm{Sc_{H2Oa}})} \tag{16}
$$

$$
\mathrm{MTC_w} = 0.01 \left( 9 \times 10^{-4} + 7.2 \times 10^{-6} U_{10}^3 \right) \left( \frac{\mathrm{MolWt_{CO_2}}}{\mathrm{MolWt}} \right)^{1/4}. \tag{17}
$$

Here, $\mathrm{MolWt}$ [g mol$^{-1}$] is the molecular weight of the target chemical, $\mathrm{MolWt_{CO_2}}$ is the molecular weight of CO$_2$ (44 g mol$^{-1}$), $\mathrm{MolWt_{H_2O}}$ is the molecular weight of water (18 g mol$^{-1}$), $U_{10}$ is the wind speed ten meters above the water surface [m s$^{-1}$], and $\mathrm{Sca_{H_2O}}$ is the Schmidt Number of water vapor in air (0.62). Then, Eqs. (16) and (17) are inserted in (15) where the dependencies on temperature and salinity are accounted for when the dimensionless Henry's law volatility constant $H^{\mathrm{cc}}$ is calculated (Park and Clough, 2014) as:

$$
H^{\mathrm{cc}} = \frac{H^{\mathrm{pc}} (1 + 0.01143\,S)}{R\,T}. \tag{18}
$$

Here, $R$ the is universal gas constant ($8.206 \times 10^{-5}$ atm m$^3$ mol$^{-1}$ K$^{-1}$), $T$ is the temperature [K] and $S$ is the salinity [PSU]. The last term at the numerator is a dimensionless linear correction factor for the effect of salinity, accounting for a factor of 1.4 for $S = 35$ PSU compared to fresh water (Park and Clough, 2014; Schwarzenbach et al., 1993). $H^{\mathrm{pc}}$ is Henry's volatility constant expressed in [atm m$^3$ mol$^{-1}$], which is either specified by the user or estimated by

$$
H^{\mathrm{pc}} = \frac{P^{\mathrm{sat}}(T_{\mathrm{ref,vp}})\,\mathrm{TempCorr}(T, T_{\mathrm{ref,vp}}, \Delta H_{\mathrm{vol}})}{C_W^{\mathrm{sat}}(T_{\mathrm{ref,sol}})\,\mathrm{TempCorr}(T, T_{\mathrm{ref,sol}}, \Delta H_{\mathrm{sol}})} \mathrm{MolWt}, \tag{19}
$$

where $P^{\mathrm{sat}}$ [atm] is the vapor pressure of the target chemical at reference temperature, $C^{\mathrm{sat}}$ [g m$^{-3}$] is the solubility of the target chemical at reference temperature, and $\mathrm{TempCorr}(\cdot, \cdot, \cdot)$ are dimensionless temperature correction factors calculated with Eq. (10) for volatilization and solubilization reactions with the respective enthalpy changes $\Delta H_{\mathrm{vol}}$ and $\Delta H_{\mathrm{sol}}$ at the target temperature $T$.

## 3 Emissions, chemical parameters and metocean forcing

### 3.1 STEAM

Emission fields from the Ship Traffic Emission Assessment Model (STEAM) by Jalkanen et al. (2021) are used in this work. Modelled emissions from open-loop scrubbers in 2019 are utilized in the following examples. The method named





seed_from_STEAM(...) is implemented in the ChemicalDrift class to allow seamless seeding of Lagrangian elements directly
from the STEAM DataArray. Emission data is provided as volume [m³] of discharged scrubber water in each spatio-temporal
grid point. The conversion from volume to mass of chemicals is implemented within the class method, applying average emissions
factors [μg/L] and confidence intervals reported in Lunde Hermansson et al. (2021) for open_loop and closed_loop
scrubbers, for a set of heavy metals and PAHs, selected with the arguments scrubber_type, and chemical_compound, respectively. The total mass of chemical for each STEAM data point is then distributed on a number of Lagrangian elements with a
given mass and over a circular area defined by the arguments mass_element_ug [μg] and radius [m].

### 3.2 Chemical parameters for PAHs

A database of parameters for a set of organic compounds and heavy metals has been compiled under the scope of the EMERGE
project, based on an extensive review of data available in the literature. Mean values from the database for a selection of
PAHs are integrated in the ChemicalDrift class method init_chemical_compound which allows to configure the whole set
of parameters by simply selecting the target chemical. Alternatively, each parameter can be specified singularly with the
set_config method. Both procedures are shown in the example in Listing 1, while the data for the PAHs utilized in this work is
reported in Table 2.

### 3.3 Metocean forcing from CMEMS and SHYFEM

Meteorological and oceanographic forcing is obtained from different sources. Several CMEMS products are utilized in this
work, providing water temperature, salinity, current horizontal velocities, mixed layer depth, ocean surface winds, and SPM
concentration, as summarized in Table 3.

High-resolution 3D currents, water temperature and salinity forcing over the Northern Adriatic Sea including the Venice
lagoon are provided by the application of the unstructured hydrodynamic SHYFEM model (Bellafiore et al., 2018). SHYFEM
model integrations are performed on a spatial domain covering the whole Adriatic Sea and its main coastal lagoons (Marano-
Grado, Venice, and the Po delta). To adequately resolve the river-sea continuum, the unstructured grid also includes the lower
part of the major rivers flowing into the Adriatic Sea. The numerical grid consists in approximately 110,000 triangular elements
with a resolution that varies from 5 km in the open sea to a few hundred meters along the coast and tens of meters in the inner
lagoon channels (Ferrarin et al., 2019).

The SHYFEM simulations are forced by atmospheric fields from the global ECMWF reanalysis ERA5 (Hersbach et al.,
2020), sea boundary and initial conditions from the Mediterranean Forecast System reanalysis (Tonani et al., 2009) and observed or climatological water discharges at main river mouths.

## 4 Application examples

The ChemicalDrift functionalities are demonstrated through a few modelling examples described in the following. The presented simulations are not meant to provide conclusive quantitative results as some of the input parameters are uncertain and





**Table 2.** Chemical parameters utilized for modelling the sorption, degradation, and volatilization of three organic compounds.

| Parameter | Tag | Naphthalene | Phenanthrene | Benzo(a)pyrene | Unit |
|---|---|---|---|---|---|
| $\log(K_{\mathrm{OW}})$ | LogKOW | 3.361 | 4.505 | 6.124 | L kg$^{-1}$ |
| $T_{\mathrm{ref,KOW}}$ | TrefKOW | 25 | 25 | 25 | °C |
| $\Delta H_{\mathrm{KOC,SPM}}$ | DeltaH_KOC_Sed | -24900 | 7507 | -43700 | J mol$^{-1}$ |
| $\Delta H_{\mathrm{KOC,DOC}}$ | DeltaH_KOC_DOM | -25900 | 7560 | -31280 | J mol$^{-1}$ |
| $k_{\mathrm{S}}$ | Setchenow | 0.2503 | 0.3026 | 0.171 | L mol$^{-1}$ |
| $t_{1/2\mathrm{Wtot}}$ | t12_W_tot | 224.08 | 1125.79 | 1491.42 | hours |
| $T_{\mathrm{ref,kWtot}}$ | Tref_kWt | 25 | 25 | 25 | °C |
| $\Delta H_{\mathrm{Wtot}}$ | DeltaH_kWt | 50000 | 50000 | 50000 | J mol$^{-1}$ |
| $t_{1/2\mathrm{Stot}}$ | t12_S_tot | 5012.4 | 29124.96 | 44934.76 | hours |
| $T_{\mathrm{ref,kStot}}$ | Tref_kSt | 25 | 25 | 25 | °C |
| $\Delta H_{\mathrm{Stot}}$ | DeltaH_kSt | 50000 | 50000 | 50000 | J mol$^{-1}$ |
| MolWt | MolWt | 128.1705 | 178.226 | 252.32 | g mol$^{-1}$ |
| $P^{\mathrm{sat}}$ | Vpress | 11.2 | 2.22e-2 | 1.36e-6 | Pa |
| $T_{\mathrm{ref,vp}}$ | Tref_Vpress | 25 | 25 | 25 | °C |
| $\Delta H_{\mathrm{ref,vp}}$ | DeltaH_Vpress | 55925 | 71733 | 107887 | J mol$^{-1}$ |
| $C_{\mathrm{W}}^{\mathrm{sat}}$ | Solub | 31.4 | 1.09 | 0.00229 | g m$^{-3}$ |
| $T_{\mathrm{ref,sol}}$ | Tref_Solub | 25 | 25 | 25 | °C |
| $\Delta H_{\mathrm{ref,sol}}$ | DeltaH_Solub | 25300 | 34800 | 38000 | J mol$^{-1}$ |

290 the model remain to be validated. The examples are run with OpenDrift 1.9.0 (e.g.) and may not be functional in the long future.

### 4.1 Organic pollutant discharge in Kattegat

A simple example of a ChemicalDrift simulation is presented with a description of the running script. The example is also available in the gallery section in the OpenDrift reference page (opendrift.github.io), where it is continuously updated with 295 live forcing data. The simulation models the fate of a mass of an organic pollutant (phenanthrene) released outside the north coast of Denmark over a period of two days. Simulation setup is done as shown in the listing below by loading the OpenDrift modules, defining a ChemicalDrift instance, adding the necessary readers for forcing data, and configuring a set of parameters. In detail, metocean forcing data including surface winds, ocean currents, sea water temperature and salinity are provided by the Norkyst800 model, while mixed layer depth is set to a constant value of 40 m. Vertical mixing is activated and the model used 300 for the diffusivity profile is selected. Released chemicals are assumed to be 90% in the dissolved fraction, and 10% adsorbed to





**Table 3.** CMEMS products and data variables.

| | |
|---|---|
| BALTICSEA_ANALYSISFORECAST_PHY_003_006 | |
| GLOBAL_ANALYSIS_FORECAST_PHY_001_024 | |
| $T$ | sea_water_potential_temperature |
| $S$ | sea_water_salinity |
| | northward_sea_water_velocity |
| | eastward_sea_water_velocity |
| $H_{\mathrm{MLD}}$ | ocean_mixed_layer_thickness_defined_by_sigma_theta |
| WIND_GLO_WIND_L4_REP_OBSERVATIONS_012_006 | |
| | northward_wind |
| | eastward_wind |
| OCEANCOLOUR_GLO_OPTICS_L4_REP_OBSERVATIONS_009_081 | |
| $C_{\mathrm{SPM}}$ | spm |

particles. Degradation rates for phenanthrene are overridden in order to produce a clear effect in this short demo, and half-life
constants are set to 6 hours in the water column, and 12 hours in the sediment layer.

```
from opendrift.readers import reader_netCDF_CF_generic
from opendrift.models.chemicaldrift import ChemicalDrift
from opendrift.readers.reader_constant import Reader as ConstantReader

o = ChemicalDrift(loglevel=0, seed=0)

reader_norkyst = reader_netCDF_CF_generic.Reader('https://thredds.met.no/thredds/dodsC/sea/norkyst800m/1
    h/aggregate_be')
mixed_layer = ConstantReader({'ocean_mixed_layer_thickness': 40})
o.add_reader([reader_norkyst,mixed_layer])

o.set_config('drift:vertical_mixing', True)
o.set_config('vertical_mixing:diffusivitymodel', 'windspeed_Large1994')
o.set_config('chemical:particle_diameter',30.e-6)   # m
o.set_config('chemical:particle_diameter_uncertainty',5.e-6) # m
o.set_config('chemical:sediment:resuspension_critvel',0.15) # m/s
o.set_config('chemical:transformations:volatilization', True)
o.set_config('chemical:transformations:degradation', True)
o.set_config('seed:LMM_fraction',.9)
o.set_config('seed:particle_fraction',.1)
o.set_config('general:coastline_action', 'previous')

o.init_chemical_compound("Phenanthrene")
```





```
# Modify half-life times with unrealistic values for this demo
o.set_config('chemical:transformations:t12_W_tot', 6.) # hours
o.set_config('chemical:transformations:t12_S_tot', 12.) # hours
```

**Listing 1.** Chemical simulation setup and configuration.

The model is at this stage configured. The next step is to seed the Lagrangian chemical elements and run the simulation. In this example, 500 chemical elements with a mass of 2000 $\mu$g are seeded within a radius of 2000 m around the selected position and within the upper 10 m of the water column.

Simulation results are illustrated in Fig. 2 showing how the chemical is partitioned between the dissolved, adsorbed to SPM, and sediment compartments and advected in the northward direction. As expected, dissolved chemicals are mainly close to the
surface and transported horizontally at higher velocities while it can be seen that elements adsorbed to SPM travel for shorter distances at a higher depths. The distribution of mass between the three modelled compartments is shown in Fig. 3 where also the exponential decay due to the combined effect of degradation and volatilization can be observed. In detail, due to the short half-lives used in the simulation, more than 40% of the initial mass was removed during the simulated period. Two clear strong resuspension events are illustrated after approximately 8 and 34 hours, as shown by the rapid increase of the mass of
chemical associated to SPM at the expense of the sediment compartment, followed by periods with a predominant exchange in the opposite direction as elements representing suspended particles are sedimented again.

```
time = datetime(2022,3,8)
ntraj = 500
iniz = np.random.rand(ntraj) * (-10.0) # seeding the chemicals in the upper 10m
o.seed_elements(lon=10.6, lat=57.6, z=iniz, radius=2000, number=ntraj, time=time, mass=2e3)

o.run(steps=48*2, time_step=1800, time_step_output=1800)
```

**Listing 2.** Seeding elements and running simulation.

### 4.2   Naphthalene and benzo(a)pyrene discharges from open-loop scrubbers

ChemicalDrift is demonstrated using input data from the STEAM model to simulate emissions of selected PAHs (i.e, naph-
thalene and benzo(a)pyrene) from open-loop scrubbers. The simulated region include emissions for January 2019 in the area between 8 and 15 degrees east, and 53 to 60 degrees north. The simulation covers also an additional period of two months after the emission of the target chemicals ended, showing how the different chemical properties (Table 2) strongly influence their environmental fate. Results are illustrated in Figs. 4 and 5. In both figures we see that naphthalene is predominantly in the dissolved phase, while benzo(a)pyrene, that has a 3 orders of magnitude larger $K_{OW}$, is mostly adsorbed to particles. Initial
discharges along ship lanes are observed in the left panels of Fig. 4. In the figure the diameter of each Lagrangian element is reduced as its associated mass decreases, indicating the target chemical's removal by degradation and volatilization. This is clearly observed for naphthalene (top-right), while benzo(a)pyrene (bottom-right) is almost entirely preserved in the simulated period since it has much longer half-lives values and it is to a large degree adsorbed to particles and more rapidly deposited to







**Figure 2.** Simulated transport and fate of a mass of phenanthrene released outside the north coast of Denmark, showing horizontal advection (top) and vertical distribution in the water column and at sea floor (bottom). The color of the Lagrangian elements indicates the partitioning between dissolved, adsorbed to SPM, and sediments.



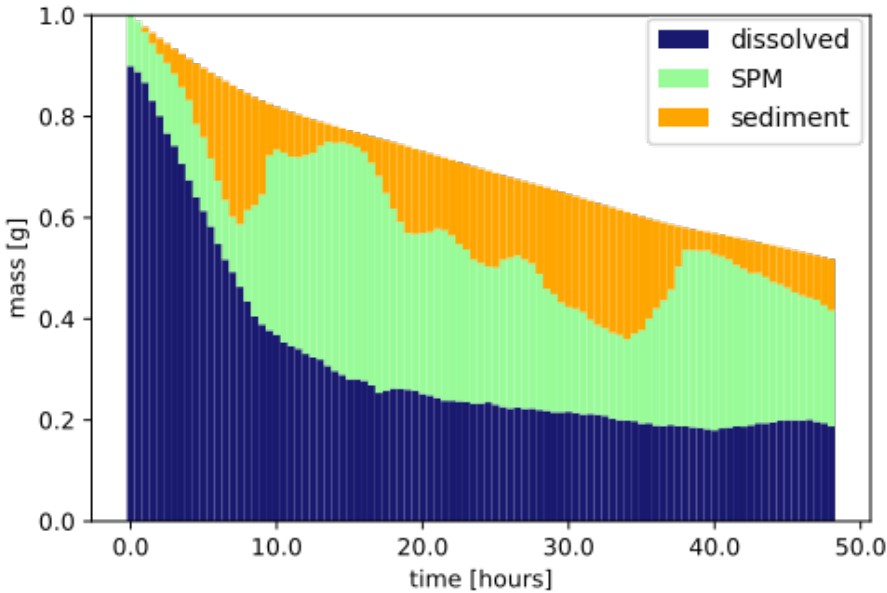

**Figure 3.** Exponential decay and mass distribution across the dissolved, adsorbed to SPM, and sediment phases for phenanthrene.

the sediment layer where its degradation is much slower. The same conclusions on the different behaviour of the two chemicals
are also observed in Fig. 5, which illustrates the two chemicals vertical profile in the water column and sediments (top), and
the time evolution and phase partitioning of the mass of chemical (bottom).

### 4.3   Multiple emission sources in the Adriatic Sea

Modelling of emissions from multiple sources in demonstrated. The target chemical considered is benzo(a)pyrene, and emissions include discharges estimated in August 2019 from open-loop scrubbers data derived from STEAM, and discharges from
49 rivers obtained from water quality monitoring data and river loads.

Snaphots of the simulation are shown in Fig.6, where colors are used to identify model compartments (top panels) and emission sources (i.e., ships or rivers, bottom panels). The use of a Lagrangian framework allows for seamless separation of the two emission sources. Emissions from scrubbers are to large degree confined in proximity of shipping lanes even at the end of the simulated period, and this is explained also considering that benzo(a)pyrene has a high $K_{OW}$ (Table. 2) and is strongly
associated to the solid fraction which is rapidly deposited to the sediment layer in proximity to the ship tracks. For the same reason, emissions from the rivers are mostly located along the shallow coastal regions, and slowly transported southwards by the cyclonic Adriatic gyres.

Simulation results are further illustrated in Fig. 7 where the mass of target chemical, and the corresponding mass removed by degradation and volatilization, are plotted versus time. The total emitted mass of chemical in August 2019 is 17500 g, and





**Figure 4.** Emissions of naphthalene (top) and benzo(a)pyrene (bottom) from open-loop scrubbers derived from STEAM model for January 2019 in the sea region around Denmark. Elements diameter is directly proportional to the associated mass. Transport and fate calculated by ChemicalDrift over 3 months period. Naphthalene is predominantly in the dissolved phase and more easily degraded and volatilized. Benzo(a)pyrene is mostly adsorbed to suspended particles and deposited to the sediment layer.



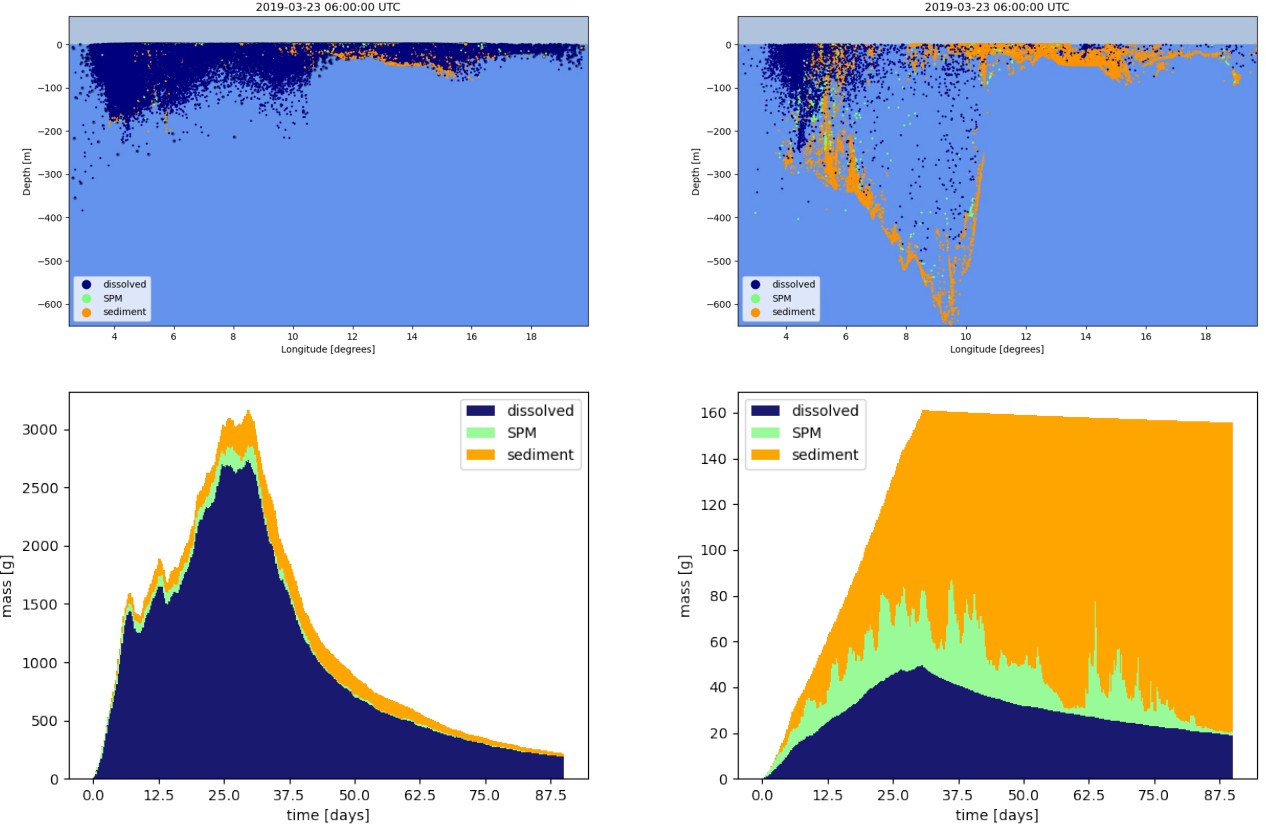

**Figure 5.** Emissions of naphthalene (left) and benzo(a)pyrene (right) from open-loop scrubbers derived from STEAM model for January 2019 in the sea region around Denmark. Northward view of the distribution of chemical in the water column and sea floor after 3 months (top), and evolution of the mass of chemicals in each model phase (bottom).

only about 500 g are removed by the end of September 2019. Degradation is dominant (470 g) compared to volatilization (30 g), which is also explained by the strong affinity of benzo(a)pyrene to the solid fraction, which to not volatilize. Most of the degraded mass is removed in the dissolved phase, even if most of the chemical is associated the sediment layer, which is explained since $t_{1/2\mathrm{Wtot}}$ is approximately 30 times shorter than $t_{1/2\mathrm{Stot}}$ (Table.2).

      The example is also utilized to illustrate the dependency on temperature and salinity of the solid-water partition coefficient 380     which is calculated according to Eqs. (8)–(10). This is reflected, e.g., on the desorption rate from the sediment to the dissolved phase, $k_{41}$ (Eq. 2), as shown in Fig. 8.





**Figure 6.** Emissions of benzo(a)pyrene from both open-loop scrubbers and rivers in the northern Adriatic Sea for August 2019, simulated in ChemicalDrift over a two-month period, showing how the use of Lagrangian modelling allows for seamless separation of the different sources. Two time steps are shown, 2019-08-08 (left) and 2019-09-29 (right). The colors indicate the phase partitioning in the top panels, and the pollutant source in the bottom panels.





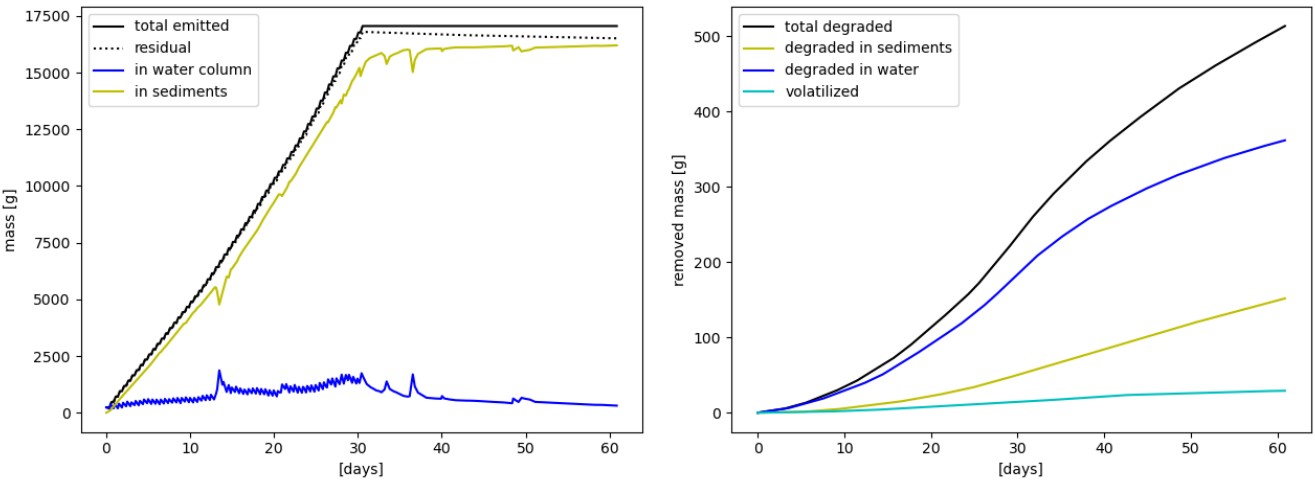

**Figure 7.** Emissions of benzo(a)pyrene from open-loop scrubbers and rivers in the Northern Adriatic Sea. Total emitted mass (left) and removed mass by degradation and volatilization (right). Mass in water column refers to all phases in the water column, including the dissolved and the SPM phase.

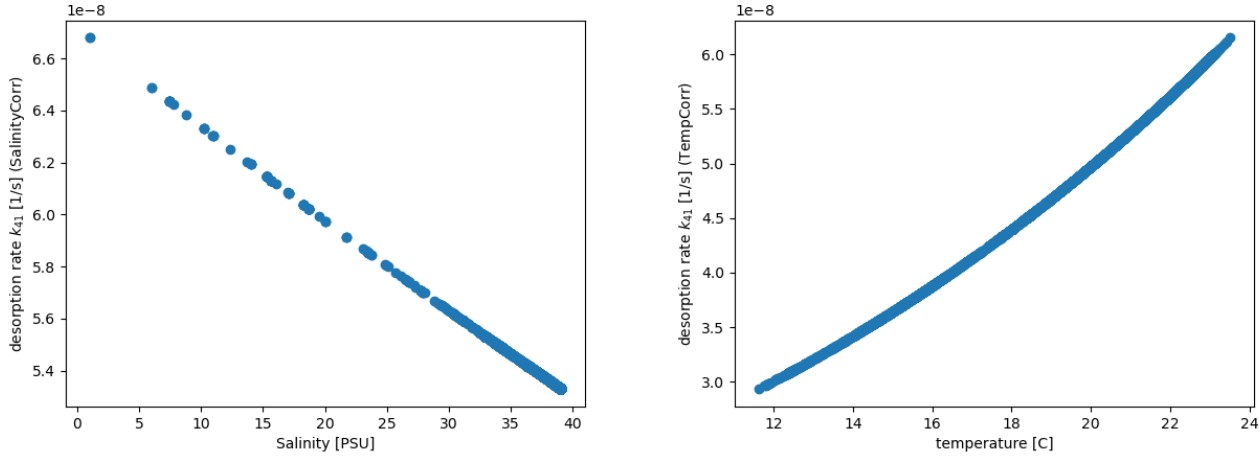

**Figure 8.** Effect on the desorption rate $k_{41}$ of benzo(a)pyrene from the sediments to the dissolved phase, due to SalinityCorr (Eq. 9, on the left) and TempCorr (Eq. 10, on the right), produced by the simulation of multiple source emissions in the Northern Adriatic Sea.





### 4.4 Modelling of open-loop scrubbers emissions in European seas

ChemicalDrift is applied to open-loop scrubbers emissions for the entire European region from January to March 2019. The target chemical is phenanthrene, and fate and transport modelling is calculated until December 2019. This example is utilized
to demonstrate the method write_netcdf_chemical_density_map(...), which is implemented in ChemicalDrift adapting the corresponding method in the Radionuclides module. The method calculates gridded concentrations of the target chemical and export these to a netCDF file. Gridding is applied as a post-processing step at the end of the simulation, providing average concentrations ($10 \times 10$ km, user definable) of the target chemical over the whole domain for each of the modelled phases. Estimated environmental concentrations are shown in Fig.9. Time step for the averaging in each grid box is also user definable.
For the modelled phases in the water column, e.g. the dissolved and adsorbed to SPM phases, it is possible to define vertical grid size, or to consider as in this case the whole column from surface to sea floor. Concentrations are saved to netCDF files. Results for the sediment phase are plotted in [$\mu g\,kg^{-1}$] where mass of chemical is the mass of Lagrangian elements deposited on the sediment layer in each grid box, and the total solid mass is calculated from user definable sediment density and active layer depth and porosity.
We see that only few ship with scrubbers were operating in the Mediterranean in 2019, since it was before global sulphur cap regulation. Dissolved chemicals are more diffused, while chemicals attached to particles sink and have smaller lateral diffusion, hence ships routes are more easily observed.

### 5 Conclusions

ChemicalDrift, a new Lagrangian model for transport and fate of chemicals in the aquatic environment, has been presented.
The model is implemented as a new module and is fully integrated within the open-source framework OpenDrift, in order to combine the newly implemented chemical processes with the framework's advanced hydrodynamical capabilities, and to provide a flexible interface with most of the available metocean input data sources.

   The modelled chemical processes include a dynamic partitioning between the different phases that pollutants can be associated to in the aquatic environment, chemical removal by degradation and volatilization, as well as sedimentation and
resuspension of chemicals associated to suspended particles. Target chemicals are modelled as Lagrangian elements that are transported and exposed to changing environmental conditions as, e.g., temperature and salinity. The dependencies of chemical processes on temperature and salinity changes are formulated and implemented.

   The focus of the presented work has been on modelling organic pollutants in the marine and coastal regions. The model functionalities are demonstrated through a sequence of simulation examples. The presented examples are only for demonstra-
tive purpose, providing insights into the combined effect of the modelled physical and chemical processes, and presenting the potential of the proposed model. Accurate tuning of input parameters, sensitivity analysis, model validation, and quantitative modelling results are deferred to future publications. Specifically, in the scope of the Horizion 2020 EMERGE project, ChemicalDrift is planned to be used for calculating concentrations of different pollutants including PAHs and heavy metals, both at the European regional scale and at finer scale in the Northern Adriatic Sea and in the Øresund strait. Additionally, the model



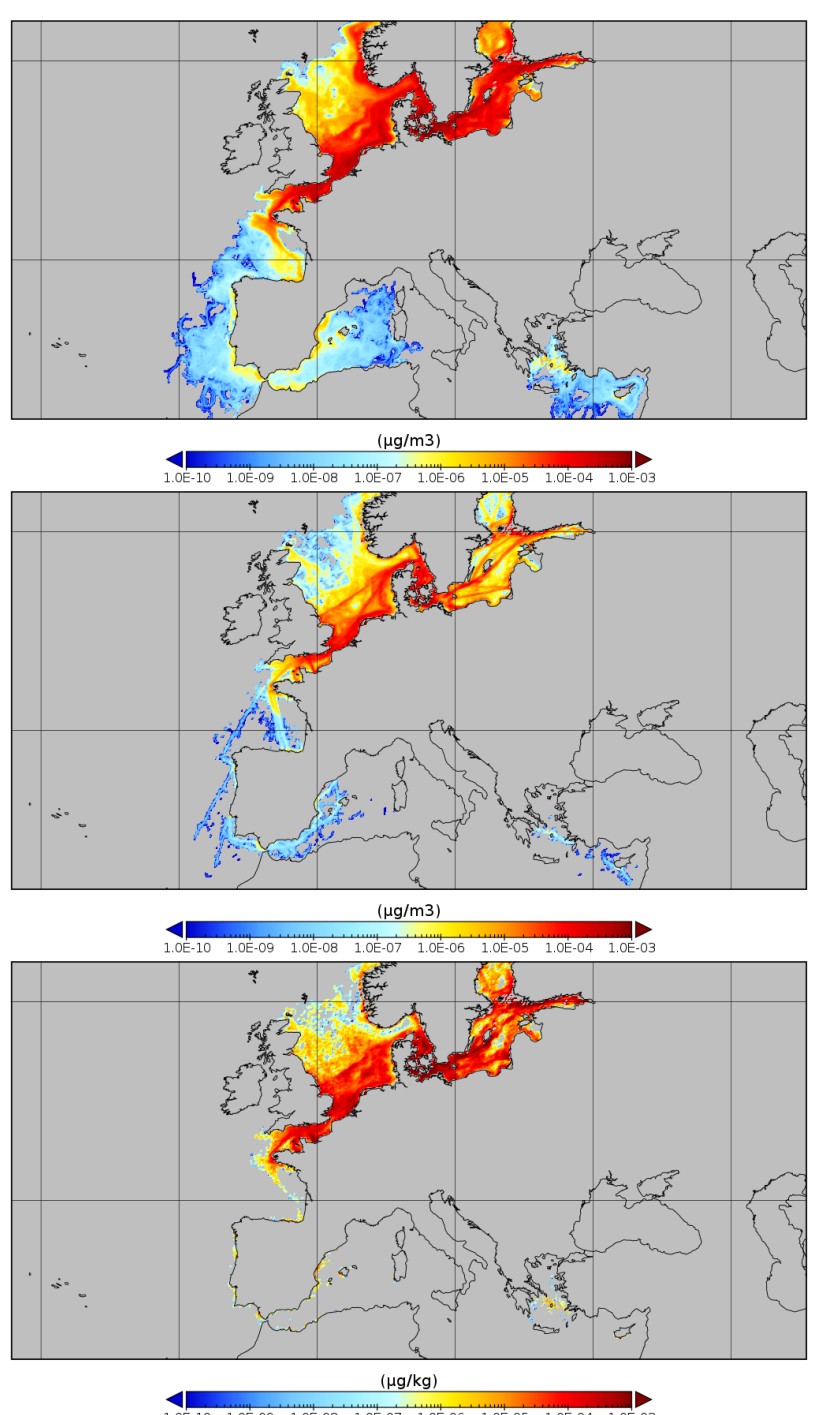

**Figure 9.** Mean concentrations of phenanthrene emitted from open-loop scrubbers (January-March 2019). Concentrations are calculated separately for the dissolved (a), adsorbed to SPM (b), and sediments fractions (c).

can be applied also to other case study areas in the EMERGE project, namely the Pireus port case study, and the Aveiro lagoon case study, where the Delft3D suite will be utilized independently by other project partners, allowing to compare the results between the two models. Datasets with measured concentrations of organic pollutants in sediments obtained from monitoring programs in the Baltic region and in the North Adriatic are identified and available and can be used in the future to attempt validation of the model.

Additional functionalities are also planned for the future, including support for hydrolysis, photolysys, and biodegradation as distinct sub-processes, support for non-ionizable organic compounds, and a more advanced resuspension scheme which will include the contribution of wave induced stress to resuspension.

High flexibility is demonstrated by the presented examples, utilizing a selection of different input sources and at different spatio-temporal scales; an interface to the STEAM model for shipping emissions, including support for open-loop and

close-loop scrubbers is implemented and demonstrated. The Lagrangian framework offers seamless tracking and separation of chemicals emitted from different sources such as emissions from shipping or from rivers. The modelled chemical processes depend on relatively large set of parameters, including e.g. the solid-water partitioning coefficient, the Henry law coefficients, and the overall degradation rates. A database of chemical parameters for a set of PAHs is integrated in ChemicalDrift, and values for three PAHs are presented.

*Code and data availability.* The current version of ChemicalDrift is freely available at the OpenDrift github repository https://github.com/OpenDrift/opendrift/blob/master/opendrift/models/chemicaldrift.py under the terms of the GNU General Public License as published by the Free Software Foundation, version 2. The exact version used for the simulations presented in this work is integrated in OpenDrift 1.9.0. The example presented in Sect. 4.1 is described with the complete running script, and available at the OpenDrift reference page (opendrift.github.io) under the gallery section including the required input data. The following examples require datasets available at Copernicus

Marine Service, except for the gridded bathymetry data that is available at GEBCO, and the STEAM and SHYFEM model results that may be made available upon request.

*Author contributions.* MA designed and developed the model, performed and analyzed the simulations, and wrote the manuscript draft. LC contributed with model design, compilation of input data, and wrote the introduction section; CF provided the SHYFEM model results and wrote the description the model in Sect. 3.3. KFD supported the development; MA, LC, LRH, ØB, KFD, CF, and AM reviewed and edited

the manuscript.

*Competing interests.* The authors declare that they have no conflict of interest.





*Acknowledgements.* This project has received funding from the European Union's Horizon 2020 research and innovation program under grant agreements no. 874990 (EMERGE project). This work reflects only the authors' view, and CINEA is not responsible for any use that may be made of the information it contains. Mattia Boscherini (Ca'Foscari University of Venice) and Isabel Hanstein (University of Heidelberg) are kindly acknowledged for the precious support in the compilation of the PAHs chemical properties database. The authors are grateful to Dr. Jukka-Pekka Jalkanen (Finnish Meteorological Institute) for providing the results of the STEAM model used in this work.






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
