# Peer review of "ChemicalDrift 1.0: an open-source Lagrangian chemical fate and transport model for organic aquatic pollutants"

_Geoscientific Model Development, 2022_

## Author Response (AR2)

**gmd-2022-212 Response to referees**

We thank the referees and the editorial board for the useful comments and suggestions. The manuscript has now been revised and resubmitted. Comments from the referees are listed below, with authors' replies, including the answer previously given in https://doi.org/10.5194/gmd-2022-212-AC2. No changes are made to the manuscript with respect to the concerns raised by anonymous referee 1.

Additionally, in the revised manuscript the code and data availability section has been updated, including the links to source code and input data archived at Zenodo. Some typos in Table 2 have been corrected.

**RC1: 'Comment on gmd-2022-212', Anonymous Referee #1, 24 Nov 2022**
**AC2: 'Reply on RC1', Manuel Aghito, 01 Dec 2022**

*Review of "ChemicalDrift 1.0: an open-source Lagrangian chemical fate and transport model for organic aquatic pollutants" by Aghito et al.*

*In this manuscript, the authors present a new modelling framework for organic pollutants, built on the OpenDrift framework. They apply the framework to three different examples, in the Baltic, North and Adriatic Seas.*

*The manuscript is generally well-written and easy to follow and navigate through. The figures are clear, and the inclusion of a code snippets gives a nice impression of the ease-of-use for the users in using the model. I can imagine that this manuscript is a good reference for new users of the model.*

*However, I do have a few very serious concerns that avoid me from recommending this manuscript for publication in GMD:*

*1 - There is essentially no validation of the code. How confident are the authors that the code works as intended? They mention functional testing in the abstract, but I could not find that in the manuscript body. Also, is there any unit testing?*

Yes, "functional testing" was used improperly, this is now rephrased to "simulation results demonstrating the functionalities of the model". Unit testing is extensively used in OpenDrift, and thus also covers the components of the new module chemicaldrift.py. Recently, after the submission to the manuscript, specific unit tests for some of the functionalities implemented in ChemicalDrift have been added to the OpenDrift github repository.

Regarding validation, our strategy is to describe the model in this manuscript, which does provide some simple verification of the equations, and provide extensive evaluation in follow-up papers, where we will provide simulations on regional and European scale. This approach is also suggested in the GMD guidelines for Model description papers, quoted "Where evaluation is very extensive, a separate paper focussed solely on this aspect may be submitted" (https://www.geoscientific-model-development.net/about/manuscript_types.html#item1) and would also fit very well with the main author publication plan for the PhD degree.

The general drift and mixing parameterisations in OpenDrift are addressed and validated in many papers using OpenDrift (see https://opendrift.github.io/references.html).

The modeled processes (partitioning, degradation, and volatilization) are implemented based on formulations used in other chemical fate models, like for example the Aquatox tool by the US Environmental Protection Agency, or provided by reference textbooks like Schwarzenbach,

Gschwend, Imboden - Environmental Organic Chemistry, so it is expected that these processes are calculated correctly.

The novelty is the integration of aquatic chemistry within OpenDrift, which allows to model the chemical processes at various scales and with seamless update of important parameters like temperature, salinity, mixed layer depth, SPM concentration, and to study the combined effect of ocean physics and chemistry.

We provide some simple verification that the chemical equations work as expected with the given examples, like in Fig.8, showing the expected effect of temperature and salinity on the chemical partitioning, or in Figs. 3-7, showing that chemical mass is reduced exponentially due to degradation and volatilization, and comparing the fate of two different chemicals, which is also consistent with our expectations since benzo(a)pyrene is much less volatile and has higher affinity to particles than naphthalene.

A comprehensive evaluation of the overall model is indeed very challenging. Concentrations of pollutants in the water column in marine environments are often lower than instruments limits of detection and cannot be measured. We can't validate if we don't have sufficient data. Moreover, chemicals in the environment are not released by a single point source like for example would be the case in a petroleum accident, so also the task of collecting all possible contamination sources (directly released to the sea by e.g. ships or oil platforms, or discharged from rivers, water treatment plants, factories, or deposited from the atmosphere) is very difficult in itself.

Nevertheless, ecotoxicological studies show that chemicals like PAHs and heavy metals can be harmful to marine species at very low concentrations, so it is important to develop tools for predicting the concentrations.

It would be a very long shot if we were attempting to provide a validation and then claiming that the model would be applicable in general, with different environmental conditions, input data obtained from different sources, and different scales.

It is more reasonable to perform parameter tuning, model calibration, and model evaluation in specific case studies. This is planned in ongoing activities and will be addressed in follow-up papers.

*2 - The authors also do not present any performance assessment. How does the code scale with number of elements? Can it work in parallel (OpenMP, MPI?) mode? What is the memory footprint of the additional codes, compared to the rest of OpenDrift? How is IO dealt with?*

Trajectory calculations are difficult to parallelize in general, but many sub-components (bottlenecks) of OpenDrift are parallelized using e.g. multiprocessing module.

We have actually worked with parallelization with ChemicalDrift after the submission of the manuscript. This has been implemented using Python multiprocessing library, splitting a simulation in 64 subprocesses, each applied to chemicals discharged in separate longitude intervals, and this gave a strong reduction of the simulation time. We can simulate for example open loop scrubbers emissions of a selected chemical for a whole year (2018) and for the whole European region, using approximately 900000 Lagrangian elements, in less than 2 hours on our HPC.

The IO is inherited from OpenDrift, and is based on export to CF-compliant netCDF files, and generic import (readers), see https://doi.org/10.5194/gmd-11-1405-2018, 2018.

*3 - In the examples, there is no assessment of the sensitivity of the results to choices like integration time-step, number of modeled elements, input/output-frequency etc; making it difficult for the reader to gauge how robust the results are to the user parameters.*

Sensitivity analysis is of course very important and needs to be performed. This should include physical parameters, chemical parameters, and numerical parameters like time-step and number of model elements. Given the large number of parameters and the broad range of variability and uncertainties of these, sensitivity analysis is also expected to provide different conclusions in different case studies. To give an example, while in one case study we might have accurate measurements of KOC and fOC, we know that in general these parameters are subject to huge variations in the environment. Hence, our plan has been to address sensitivity analysis as a step of model verification in follow-up papers focussing on specific case studies.

*4 - The code builds heavily on OpenDrift itself, and is in some way an obvious further extension of the Radionuclides extension. Most of the new code is simply an implementation of physical equations, presented without much numerical or computational consideration.*

We focussed on describing the novelty, which is the integration of the chemistry of organic compounds in the OpenDrift framework. This is not in OpenDrift, and not in Radionuclides. All equations described in the manuscript correspond to new code implemented in ChemicalDrift.

The numerical framework is indeed largely inherited from OpenDrift, hence the focus of this manuscript is on the chemical processes, and not the numerical implementation.

*5 - Because of the four points above, I seriously doubt that this manuscript falls within the scope of GMD. The way it is presented, it feels more like an application of OpenDrift to a few very specific chemical processes within the context of a European project; rather than a versatile and potentially widely useable community code. Perhaps another journal (e.g. on marine pollution) might be more relevant for this work?*

The chemical processes handled within ChemicalDrift are general, and not limited to the specific EMERGE project, although it provided funding for the development. The model can easily be used in many other applications where risk assessment of contamination of organic pollutants is useful: discharges from shipping in general, fish farms, produced waters for oil platforms, discharges from rivers, water treatment plants, deep sea mining. The integration within OpenDrift, which provides a very flexible and simple interface, will facilitate the use of ChemicalDrift in other applications.

Moreover, ChemicalDrift can easily be extended to other types of chemicals, like for example ionizable compounds, and heavy metals, and updates made after the submission of the manuscript are already available on the github repository.

The examples given in the paper are only preliminary demonstrations, presented without any quantitative analysis, for the scope of describing the model. The examples are based on EMERGE data since this is the project we have been working on. We think that follow-up papers presenting and analyzing simulation results might indeed be more suited to another journal with more focus on environmental chemistry.

*Furthermore, I also have some minor comments and suggestions:*

*lines 8, 18, 221, 423 and other locations: At places, words like 'powerful', 'valuable', etc make it sound more like a sales-pitch than a self-critical scientific assessment. I suggest to be very careful with this self-congratulatory framing.*
(line 8): "powerful" changed to "flexible"
(line 19): "valuable" changed to "freely available and open-source"
(line 225): removed "reasonable"

*line 26: Can it be assumed that all readers know what a Lagrangian model is? Also, OpenDrift could be explained in more depth*
(line 27) Reference to a recent review paper on Lagrangian analysis is added.

*line 66: what is a 'severe' oil spill? Why would it not work for non-severe oil spills?*
(line 66) Removed 'severe' which might be confusing.

*line 67: this wording suggests that the only difference between OilDrift and ChemicalDrift is the concentrations that can be tracked; but I think there are many more differences? Perhaps rephrase?*
(lines 68-71) Yes there are many more differences. The sentence has been rephrased and should be more clear.

*line 73: mention here already that (re-)deposition is not taken into account?*
Re-deposition is taken into account. After deposited sediments return to the water column either by dissolution or re-suspension, these can later be re-deposited. This typically leads to sediments stepwise accumulating in deeper spots.

*line 74: I don't think this statement is meant to imply that only findings from open-access literature is used (so ignoring closed-access literature)? Perhaps rephrase?*
(line 77) Right, Removed "open".

*line 85. Fig 1 caption and further: I am a bit confused whether the terms 'compartment' and 'phase' refer to the same concept, or something different. If they are the same, I suggest using only one of the terms throughout*
Changed "Optional phases" to "Optional compartments" in Fig.1 caption.

*line 168: 'neglected' is not the right word here. Vertical velocity is a diagnostic variable in most ocean models*
(line 171) Changed "typically neglected in operational ocean models" to "typically small compared to the other terms".

*line 172-174: How good is this assumption here? What is the typical error made when using this assumption in realistic scenarios?*
This was not described accurately in the submitted manuscript. The simplification of assuming low Reynolds number is not utilized in ChemicalDrift, while
the more general algorithm for calculating particle terminal velocity is used, based the work of (Sundby, 1983) https://doi.org/10.1016/0198-0149(83)90042-0 and implemented in LADiM (https://github.com/bjornaa/ladim1) and other OpenDrift modules (larvalfish, pelagicegg).
(line 175) Removed ", and low Reynolds numbers" and added reference. Particle size and density have a significant impact and must be properly selected in each specific scenario.

*line 175-179: what is the physical interpretation and motivation behind this parameterization?*
(line 179-183) A reference is added for this simple parameterization. Specified that we refer to the current at the bottom. More complex schemes keeping also into account the particles size and density, as well as the stress induced by surface waves (in shallow water), might be implemented in the future, as also mentioned in the conclusions.

*line 226: k should be type-set in mathmode?*
(line 230) Corrected.

*Table 3: I'm a bit surprised that the authors use the interpolated CMEMS data here. Why not the non-interpolated original version of PHY on the MOi servers at e.g. https://www.mercator-ocean.eu/en/solutions-expertise/accessing-digital-data/product-details/?offer=42 17979b-2662-329a-907c-602fdc69c3a3&system=d35404e4-40d3-59d6-3608-581c9495d86a (which also includes vertical velocity)*

This is a useful suggestion that we can consider in future works. OpenDrift (and therefore ChemicalDrift) can use vertical current velocities if these are provided. The simulations provided in this work are only for demonstrative purposes, the focus is not on using the most accurate and complete input data.

*line 355: while helpful for visualisation, I fear that such a representation of element 'mass' by size might be a bit misleading to some readers, who may not appreciate that elements don't actually have an actual 'size'.*

(line 360) It is true, this representation might be misleading in some cases. It is very useful for presenting and analyzing the model. It does in many situations provide a more realistic impression than obtained when keeping the size of the Lagrangian elements constant despite the fact that the actual mass is varying. The user can select the initial size, and the elements are plotted with some transparency, so that areas where Lagrangian elements accumulate will be shown with more intense colors. The post-processing tool to calculate actual concentrations of the target chemical (sect. 4.4, Fig.9) gives indeed the most objective output.

*line 363: 'is demonstrated'*
(line 367) Corrected ("in demonstrated" changed to "is demonstrated").

*line 367: The statement about the Lagrangian framework here is a bit misleading. It would be perfectly possibly to run two separate tracers in an Eulerian framework and that way also separate the sources. The Lagrangian framework does have advantages, but the way it's formulated now is not one of them*
(line 371) Removed "The use of a Lagrangian framework allows for seamless separation of the two emission sources."

**RC2: 'Comment on gmd-2022-212', Anonymous Referee #2, 17 Feb 2023**

*The authors present ChemicalDrift, a new module of the Lagrangian framework OpenDrift. The background and motivation for this new capability is clearly introduced. The chemical modelling is then comprehensively outlined, covering key equations and parameters, including some physical state (temperature and salinity) dependence. Chemicals of interest are allocated to dissolved, suspended particulate and in-sediment states. A necessary range of physical and chemical processes are parameterized, including adsorption and desorption, sedimentation and resuspension, degradation, and volatilization.*

*ChemicalDrift may be driven with source emissions determined with Ship Traffic Emission Assessment Model (STEAM) and/or with riverine inputs. Subsequently subject to variable hydrography, currents and winds, the module is used in a series of examples: an organic pollutant released into the Kattegat and traced for 34 hours; two types of emission from STEAM are traced in a wider region encompassing Baltic and North seas through 2019; the fate of benzo(a)pyrene from both shipping and rivers in the northern Adriatic over 2 months; fractional dispersal of phenanthrene emitted from shipping (open-loop scrubbers) across European seas in during January-March 2019.*

*In conclusion, the authors make clear the plans for development and validation of ChemicalDrift. Capable of high-fidelity source-to-fate tracing of pollutants from multiple sources, ChemicalDrift is a flexible new tool for the marine science community, with potential for widespread application. As a*

*model description paper, the manuscript should be suitable for publication in GMD, subject to technical corrections as indicated below.*

*Technical Corrections:*

*1 caption: do you mean 'wavy lines' rather than 'dashed lines'?*
Changed "solid lines" and "dashed lines" to "solid contours" and "dashed contours" which should be more clear since we are referring to the boxes, not to the arrows.

*Lines 168-169: reads better as 'associated with', rather than 'to the effect of'*
(172) Rephrased.

*Fig 4 caption: You explain 'Elements diameter is directly proportional to the associated mass.'; this is clear in the upper panels, but diameter seems constant in the lower panels; is this correct?*
Degradation and volatilization are much slower for benzo(a)pyrene (bottom panels), therefore no significant change is observed in this case. The slow removal of benzo(a)pyrene is more easily observed with the total mass balance shown in Fig.5.

*Line 376: 'which does not volatilize' rather than 'which to not volatilize'*
(379) Corrected.

*Line 387: 'exports' (rather than 'export')*
(390) Corrected.